# How Do Nursing Students Perceive Moral Distress? An Interpretative Phenomenological Study

Chiara Gandossi [1], Elvira Luana De Brasi [1], Debora Rosa [2,*], Sara Maffioli [1], Sara Zappa [1], Giulia Villa [2] and Duilio Fiorenzo Manara [2]

[1] San Raffaele Institute (IRCCS), 20132 Milan, Italy
[2] Center for Nursing Research and Innovation, Vita-Salute San Raffaele University, 20132 Milan, Italy
[*] Correspondence: rosa.debora@unisr.it

**Abstract:** Background: Research shows that the longer nurses care for terminally ill patients, the greater they experience moral distress. The same applies to nursing students. This study aims to analyze episodes of moral distress experienced by nursing students during end-of-life care of onco-hematologic patients in hospital settings. Methods: This study was conducted in the interpretative paradigm using a hermeneutic phenomenological approach and data were analyzed following the principles of the Interpretative Phenomenological Analysis. Results: Seventeen participants were included in the study. The research team identified eight themes: causes of moral distress; factors that worsen or influence the experience of moral distress; feelings and emotions in morally distressing events; morally distressing events and consultation; strategies to cope with moral distress; recovering from morally distressing events; end-of-life accompaniment; internship clinical training, and nursing curriculum. Conclusions: Moral distress is often related to poor communication or lack of communication between health care professionals and patients or relatives and to the inability to satisfy patients' last needs and wants. Further studies are necessary to examine the quantitative dimension of moral distress in nursing students. Students frequently experience moral distress in the onco-hematological setting.

**Keywords:** moral distress; nursing education; psychological stress; undergraduate nursing students





## 1. Introduction

Nurses experience moral distress (M.D.) when they feel that the ethically correct action to take is different from what they are tasked with doing [1]. Several studies are focused on moral distress among the nursing population [2–5]. Research shows that the longer nurses care for terminally ill patients, the greater they experience M.D. [6–8].

In 1984, Jameton defined M.D. as "a condition that occurs when the nurse makes a judgement by virtue of a given situation and is faced with obstacles that prevent the next action to be taken" [9]. Subsequently, the same author distinguished M.D. into initial and reactive. In the former, the nurse experiences frustration, anger and anxiety in the face of obstacles to instructions and interpersonal conflicts over professional values [1]. In the second, the nurse experiences the consequences of having difficulty in processing feelings arising from the initial MD. Other authors have expanded the concept of MD and introduced the concept of moral residue [10]. These authors define moral residue as persistent feelings and personal disagreements resulting from the MD that continue even after the event has ended. It can cause much damage over time, especially when a person is repeatedly exposed to morally distressing events. Nurses may be exposed to a 'crescendo effect' phenomenon [11]. Indeed, repeated exposure that accumulates over time can affect the moral conscience of professionals, causing great distress and putting future actions at risk. The same applies to nursing students [12–14]. Nursing students in clinical settings improve their own learning, performance, and autonomy [15]. Despite

this, internship clinical training can also lead to ethical conflict and dilemmas that could deprive students of their capability in nursing care. Research shows that morally distressing events are first experienced by nursing students in nursing care offered to terminally ill patients [16,17]. Only a few papers highlight the impact of nursing care on nursing students in onco-hematological settings. In the nursing curriculum, a goal of end-of-life nursing education is to train nurses who are comfortable with death and dying [16]. Likewise, in internship clinical training, the nursing assistant plays an important role [18]. Nursing students' experiences of morally distressing events can have important implications; some of them reported experiences of a sense of failure, disappointment, anger, guilt, nervousness, confusion, and frustration [19]. Moreover, M.D. can cause anxiety and physical symptoms such as gastrointestinal issues, insomnia, and headaches [18,20]. This study aims to analyze the episodes of M.D. experienced by nursing students during the end-of-life care of onco-hematologic patients in hospital settings, to describe their involvement in decision-making, their coping strategies, consequent reflections, and the effects on their profession and future career.

## 2. Materials and Methods

### 2.1. Study Design

This study was conducted in the interpretative paradigm using a hermeneutic phenomenological approach [21].

### 2.2. Recruitment and Sampling

Purposeful sampling was used to select meaningful information about participants for a detailed study [22,23]. Researchers located second and third-year nursing students with experience in onco-hematological settings. No deliberate selection based on demographic characteristics was made. The snowballing sampling method was then applied [24]. The researchers contacted each student by telephone, explaining the aim of the study and gave an appointment to the students who decided to participate in the study. Each participant was assigned a numeric code to ensure anonymity and to allow researchers to identify and contact the student easily whenever they needed clarifications or explanations.

### 2.3. Data Collection

The study was conducted following the consolidated criteria for reporting a qualitative research checklist (COREQ) [25]. The interviews were conducted between March 2017 and August 2018. The interviews were conducted by a female PhD nurse researcher who is an expert in qualitative research and who was not involved in nursing training. They had an informal talk with the students to facilitate the interaction and let participants tell their own stories, in their own words [24,26]. At the end of each interview, the researcher took notes of the patient's or interviewer's emotions, body language, and interjections to have a complete picture of what had happened. The research team consisted of four undergraduate students from the Bachelor of Science in Nursing course, one PhD in Nursing and Public Health, and an associate professor in Nursing. The interviews took place in a quiet and secluded location, for example, in the place where the phenomenon under investigation routinely happens [27]. Data were collected through in-depth face-to-face interviews [24]. The interviews were made up of open-ended questions about morally distressing events experienced by students and probing questions when needed (Table 1) [28]. Sample size and domain size were estimated at the point of saturation [24].

**Table 1.** The qualitative interview.

| Italian Question | English Question |
|---|---|
| • Sai che vogliamo studiare reali esperienze di stress morale legate a specifici fatti ai quali hai assistito. Ti è mai capitata una situazione clinica con un paziente morente, con i suoi parenti o con i colleghi che ti ha provocato uno stress morale?<br>• Te la senti di raccontarcela? | • You know we want to study real experiences of moral distress related to specific facts you witnessed. Have you ever had a clinical situation with a dying patient and his relatives or colleagues that caused you moral distress?<br>• Do you feel like telling us about it? |
| • Ti sei sentito coinvolto nelle decisioni prese in quella situazione?<br>• Che cosa hai fatto?<br>• Come hai affrontato questa situazione? (quali strategie hai adottato: azioni, modi di essere e modi di reagire)<br>• Ti ricordi cosa pensavi in quei momenti? | • Did you feel involved in the decisions you made in that situation?<br>• What did you do?<br>• How did you deal with this situation? (What strategies you adopted: actions, behaviors and reactions)<br>• Do you remember what you were thinking at the time? |
| • Quali emozioni/sentimenti hai provato durante e dopo?<br>• Per quanto tempo hai continuato a pensarci? | • What emotions did you feel during and afterwards?<br>• How long did you keep thinking about it? |
| • Hai parlato con qualcuno di questa esperienza?<br>• Hai avuto sostegno dai colleghi\coordinatori e dagli assistenti di tirocinio\tutor universitari?<br>• Hai percepito il contesto di tirocinio come supportivo in questa esperienza di stress morale?<br>• Hai riscontrato la presenza di risorse e/o strategie di supporto agli infermieri che affrontano esperienze di stress morale?<br>• Che cosa è successo al paziente/parenti in seguito a questa situazione?<br>• Che cosa è cambiato in te dopo quell'esperienza? | • Have you talked to anyone about this experience?<br>• Did you get support from nurses, doctors, training assistants, academic tutors, or other interns?<br>• Did you perceive the internship context as supportive in this experience of moral distress?<br>• Did you notice the presence of resources and/or support strategies for nurses facing experiences of moral distress?<br>• What happened to the patient/patients as a result of this situation?<br>• How did you change after that experience? |

*2.4. Data Analysis*

For the data analysis, the researcher did not use any software. Data were analyzed following the principles of the Interpretative Phenomenological Analysis (IPA) [24]. The first step of the IPA provides the immersion of each researcher in the original data. Two researchers independently started the data analysis, investigating one transcript after another in-depth. At this stage, it is important to read and re-read the transcripts, make margin notes, create a summary list of the margin notes, develop the emergent themes, and look for connections among them. In cases of disagreement, the two researchers returned to the original texts of the interviews and their notes and reformulated the shared themes. Once each interview is individually analyzed, the process is then extended to scan the entire set of transcripts for a full listing of themed summaries, a grouping of the themed summaries, recoding transcripts with overall themes, and finalizing the list of themes with extracts [24].

*2.5. Study Rigor*

The criteria to promote trustworthiness referred to credibility, transferability, dependability, and confirmability [29,30]. Strategies have been adopted to ensure trustworthiness: the prolonged engagement of the researchers in the study, multiple interviews and notes, member checking, and triangulation [29,30]. Verbatim quotes were included throughout the analysis to give the participants a voice, so that readers could trace back the research team's interpretations, demonstrating sensitivity toward the context [31].

*2.6. Ethical Consideration*

The study was developed per the Helsinki Declaration and the national ethical principles for scientific research and obtained the approval of the centers involved. All partici-

pants were informed of the purpose and methodology of the study by signing an informed consent form. Given that the study involved nursing students alone, non-formal ethics committee approval was needed. A synopsis of the study protocol was, however, submitted to the local committee, which gave us the clearance to proceed.

## 3. Results

From a pool of 21 nursing students, 18 females and three males, three nursing students have been excluded because they had not experienced a morally distressing situation during their internship, and one nursing student because he did not give his consent (Table 2). Seventeen nursing students agreed to participate, fifteen women, and two men with a mean age of 22.1 years. Sociodemographic data were shown in Table 2. These interviews were audio-recorded and transcribed verbatim. The interviews lasted between 22′54″ and 107′24″ with a total mean recorded time of 53′13″.

**Table 2.** The demographic characteristics.

| | | |
|---|---|---|
| Age (mean) | 22.1 | |
| Gender | Female | 15 (88.2%) |
| | Male | 2 (11.8%) |
| Internship clinical training in onco-hematological setting | Second year | 11 (64.7%) |
| | Third year | 6 (35.3%) |
| University lectures about nursing in the end-of life care in the second year of nursing school | Yes, before the internship clinical training | 7 (63.6%) |
| | Yes, after the internship clinical training | 4 (36.34%) |
| University lectures about ethics and moral philosophy in the third year of nursing school | Yes, before the internship clinical training | 3 (50%) |
| | Yes, after the internship clinical training | 3 (50%) |

There was consensus among the participants in the in-depth interviews that M.D. was a common experience in their internship clinical training. 953 units were identified from transcribed interviews and clustered into 368 labels, 47 categories, and eight themes (Figure 1).

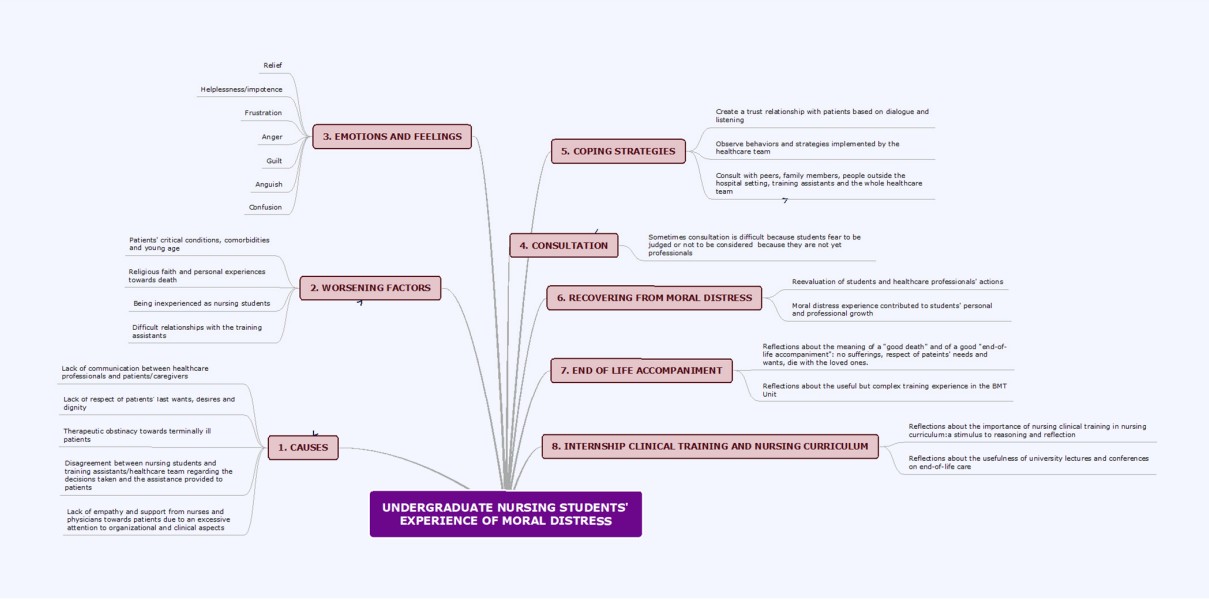

**Figure 1.** Concept map.

*3.1. Causes of M.D.*

M.D. is often related to poor communication or a lack of communication between health care professionals and patients and relatives or both, and to the inability to satisfy patients' last needs and wants. Family members' attitudes often influenced patients' end-of-life treatment decisions. In fact, in many situations relatives exerted considerable pressure on patients in various ways, sometimes more than they intended to. However, these situations are recognized as a source of M.D. by nursing students.

"[In my opinion, the decision to undergo a bone marrow transplant] seems [to have] been made by other people [rather than the patient]. It seems that it is the [patient's] relatives who want this more than himself. Since my first day of internship clinical training, the wife has had the fixed idea of the [bone marrow transplant] into her head and [she used to say]: "Ok, you'll undergo the bone marrow transplant". I never saw an exchange between her and patient. The relatives [urged the patient] to get up, settle down even on days when you could see he was really tired. His wife was always trying to force him to eat, even when [it was obvious that the patient was suffering]." (13.S.18, U.S. 13.6, 13.7)

Several nursing students experienced M.D. in situations in which they felt that medication was given only to satisfy a medical order or relatives' wants. Moreover, the participants described situations in which they felt that starting a new therapy was totally inappropriate for the patient's health status and not necessary to alleviate their suffering. Differences in the assessment of the patient's condition between students and their nursing assistant or physician were also mentioned as morally distressing.

"I experienced M.D. in the onco-hematology department when I disagreed with a physician's decision [related to a transplant] on a terminally ill patient who had other comorbidities [and therefore had an unfavorable primary condition for transplant]". (12.S.18, U.S. 12.1)

Therefore, nursing students experience M.D. when a patient or family member or both, are unaware of the terminal clinical condition due to the lack of clear communication by physicians.

*3.2. Factors That Worsen or Influence the Experience of M.D.*

Some clinical conditions (like frequent relapses of the disease, severe co-morbidities, or the young age of the terminally ill onco-hematological patients) are factors that have negatively influenced the nursing students' experience of M.D. Besides nursing students' lack of experience due to their status and the difficult relationship with their training assistants and the health care professionals, were factors that made the experience of M.D. even worse.

"It was the first time that I was dealing with a dead patient and her relatives. I was petrified [ . . . ] I was a spectator and at that very moment I didn't act, I didn't take charge of the situation, I didn't say anything, I just stood there and watched . . . I felt emotionally unstable . . . I didn't know how to behave . . . I didn't know whether to go [away] or stay there and say nothing or leave [the dead patient and her husband] alone for a moment". (06.S.18, U.S. 6.61)

In some interviews, students pointed out that their religious faith and personal experiences (such as the death of a relative because of cancer) had sometimes influenced their perception of M.D. However, the M.D. of the students was relieved when their point of view on a situation coincided with that of the care team, making them feel supported and safe. Often, students cope positively with morally distressing events when the training assistant and the whole care team share their thoughts and caring actions.

### 3.3. Feelings and Emotions in Morally Distressing Events

Nursing students have contrasting feelings and emotions when caring for terminally ill patients. Only a few nursing students experienced positive emotions (related to unexpected healing). Morally distressing events led most nursing students to experience feelings of powerlessness, anger, anguish, and frustration in assisting terminally ill patients and, for this reason, many of them realized they needed to talk to other people and think about their experience, wondering how they could have acted differently.

"[In this situation I was feeling] so much sadness, I was feeling like crying and I was feeling dizzy. I had seen [patients] in acute situations or before or after death, but I had never lived the moment of death together with the patient". (03.S.18, U.S. 3.94)

### 3.4. Morally Distressing Events and Consultation

Many nursing students highlighted that the consultation after a morally distressing event was useful. After the consultation with an internship assistant, they felt supported and understood and it helped them to better deal with their malaise.

"The fact that I talked about it with my assistant and my colleagues is a strategy [that I adopted to deal with my discomfort due to lack of communication]. Telling the story was a way to share the situation and not [bear it] alone. It was also useful to collect [different] opinions and see if that [situation] only bothered me or also others". (05.S.18, U.S. 5.42)

On the other hand, other participants reported that the consultation did not allow them to resolve their ethical dilemma because neither their colleagues nor the training assistants showed sympathy. Some nursing students finally preferred not to consult anyone due to a lack of courage or the fear of being judged.

### 3.5. Strategies to Cope with M.D.

Several strategies to cope with M.D. were reported by nursing students who realized that students, registered nurses, and patients' relatives had different ways to deal with it.

Nursing students sometimes prefer to remain silent and just listen to the patient to avoid saying something inappropriate. In other cases, they distracted the patient by trying not to talk about the illness.

"One strategy for dealing with the problem [was] to talk [with the patient], to distract him [by making him think about the things he liked], to make him tell me the personal episodes that I had heard from others, instead of talking about the [transplant, as he often did]". (03.S.18, U.S. 3.51)

Nursing students observed that some registered nurses who were accustomed to dealing with death seemed cold and a little standoffish even though they realized that their attitude was not due to a lack of sensitivity but to the attempt to avoid being involved emotionally.

"Nurses are used to these situations because they live them every day and, in front of this situation, I found them quite cold. Maybe it's a matter of habit . . . living [these situations related to terminally ill patients] every day creates a sort of indifference". (10.S.18, U.S. 10.31)

### 3.6. Recovering from Morally Distressing Events

Many nursing students re-thought and re-evaluated morally distressing events. They wondered if they had done their best and made the right choice. This process allowed them to re-evaluate their actions and professionals' actions, wondering how they would have acted if they had been in their place.

The experience of an M.D. had sometimes led to students' professional and personal growth.

"I think, for better or worse, the episode with this patient has been constructive for my professional and personal growth . . . if I can remember it even months later, it means that it has caused a really strong reaction in me. [ . . . ] I'm more and more convinced of the university choice I made, and I think that being a nurse is the right thing for me and the fact that I've established this relationship with her cheers me up." (02.S.18, U.S. 2.48, 2.50)

### *3.7. End-of-Life Accompaniment*

Some nursing students discussed the concept of "good death" and reflected on what a "good" end-of-life accompaniment should be like. According to students, a good end-of-life accompaniment should guarantee pain management and the satisfaction of patients' last needs and wants. Moreover, the closeness of relatives could be useful to terminally ill patients.

"I don't think anyone can say what a patient's end-of- life should be like specifically. But, in my opinion, when it is known that [the patient's death] is imminent, health care professionals should try to relieve patient's sufferings, to console him, to ask him what he wants to do to relieve [his] sufferings and to make him feel better, to help him understand what will happen so that he will have a conscious and peaceful death, [without therapeutic obstinacy]. (12.18, U.S. 12.15)"

### *3.8. Internship Clinical Training and Nursing Curriculum*

According to the participants in this research, university lectures on end-of-life care ensure a good theoretical preparation to deal with situations of M.D. even if, only during clinical training, students can apply the theory, deepen it, and sometimes even better understand it. Some nursing students also highlighted that they have deepened the issue of end-of-life care within conferences or educational opportunities outside the university context.

## 4. Discussion

The aim of this study was to analyze the episodes of M.D. experienced by nursing students during the end-of-life care of onco-hematology patients in a hospital setting, to describe their involvement in decision-making, coping strategies, consequent reflections, and the effects on their profession and future career.

Onco-hematology, as well as other care settings investigated in the literature [32–34], is a care setting in which nursing students often experience M.D. due to its ethical, care, relational, and organizational complexity. The interviewed students pointed out that M.D.s are often generated by reduced or absent communication between caregivers and patients and relatives or both, and the inability to meet patients' ultimate needs and wishes. Furthermore, the respondents highlighted the presence of conflicting demands from relatives [2,16,18,35,36]. Communication is a very important soft skill for nursing professionals and university education should help students to cope with the daily challenges of nursing. Developing the "soft skills" of communication through education would help students, and thus future professionals, to make the best use of the technical skills for the patient in front of them [36–38].

The students' M.D. experiences were also related to decision-making problems concerning the difference in thinking between students and training assistants and between students and the multidisciplinary team or both, with particular emphasis on the difficulty of relating to professionals who were not always willing to exchange views. Furthermore, regarding the support received from university and clinical tutors, students often stressed the difficulty of expressing their experiences for fear of not being heard or of being judged and evaluated negatively. Previous studies, however, suggest that sharing daily frustrations with peers as well as with leaders are protective factors against M.D. [39], both for students and professionals. Furthermore, positive social relationships among peers have been a widely used strategy for students, who have found comfort in peers [36]. They allow the

student an open and honest confrontation, which helps them to reflect and reframe their experiences, even individually, understanding their nuances and facets and thinking about how to deal with similar situations in the future.

The present study showed that many of the situations that generate M.D. and the strategies implemented by students are the same as those described by nurses [33,36]. The feelings and emotions related to the students' experience of M.D. were negative: feelings of helplessness, anger, anxiety, confusion, and frustration accompanied the participants even after some time, underlining the long-term impact of their experiences of discomfort. This could mean that students could become nurses who will manifest an early M.D. residual [1], resulting in a moral residue [11]. This could generate the crescendo effect, which would lead to young nurses being very dissatisfied with their work and even leaving the profession a few years after starting it [40].

Therefore, training can help to reduce, manage, and prevent M.D. also in students. Trainers, who are the facilitators of learning, should help the student to identify situations that generate M.D. and to implement strategies and interventions to manage it, and prevent negative effects on students, and thus on future professionals.

*Limitations*

In analyzing the data, the research study did not consider certain factors that could influence an individual's perception of M.D. such as gender, religious beliefs, and any previous personal experiences of participant's contact with the end of life. Furthermore, as this is a qualitative study, the results are not replicable as the subjects, experiences, and context are unique [41].

**5. Conclusions**

Further studies are necessary to examine the quantitative dimension of M.D. in nursing students, and also in different clinical settings, to identify strategies that might help students manage their lived experience during their nursing education.

**Author Contributions:** Conceptualization D.F.M. and G.V.; methodology, D.F.M. and G.V.; formal analysis, C.G., E.L.D.B., S.M. and S.Z.; writing—original draft preparation, C.G., E.L.D.B. and S.M.; writing—review and editing, D.R.; supervision, G.V. and D.F.M.; funding acquisition, D.F.M. All authors have read and agreed to the published version of the manuscript.

**Funding:** This research was funded by the Center for Nursing Research and Innovation at Vita-Salute San Raffaele, Milan, Italy. The funder did not play any role in the conduction or publication of the study.

**Institutional Review Board Statement:** The study was conducted in accordance with the Declaration of Helsinki.

**Informed Consent Statement:** Informed consent was obtained from all subjects involved in the study.

**Data Availability Statement:** The data presented in this study are available on request from the corresponding author.

**Conflicts of Interest:** The authors declare no conflict of interest.

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
