# Peer review of "How Do Nursing Students Perceive Moral Distress? An Interpretative Phenomenological Study"

_nursrep, doi:10.3390/nursrep13010049_

Round 1

Reviewer 1 Report

The number of respondents in future research should be larger if it is to answer whether and what learning outcomes are needed in teaching. I recommend the recent WHO publication: Global Competency and Outcomes Framework for Universal Health Coverage. Geneva: World Health Organization; 2022, which can support the preparation of conclusions for the practice in the future publications

Author Response

The authors thank the reviewer.

Reviewer 2 Report

First of all, I want to congratulate the authors of this study. In fact, it is a relevant topic for the academic and scientific community.

I give some observations, which should be taken into account by the authors.

Introduction – improve the theoretical framework and the object of study of the investigation.

Lines 27-28 – Are references 2-5 all from systematic reviews? Clarify please.

Lines 96-99 – Is there a reference number to the Ethics Committee authorization?

Materials and Methods – the use of resources such as the Consolidated criteria for reporting qualitative research (COREQ): a 32-item checklist for interviews and focus groups is not described. What is the reason?

Discussion – it is suggested a greater depth in the discussion, even to support the Conclusion of the study.

Author Response

Introduction – improve the theoretical framework and the object of study of the investigation.

Thank you for your comment. We have added a theoretical framework in the introduction.

Lines 27-28 – Are references 2-5 all from systematic reviews? Clarify please.

Thank you for your comment. We have changed the word “systematic”

Lines 96-99 – Is there a reference number to the Ethics Committee authorization?

Thank you for your comment. Given that the study involved nursing student alone, non-formal ethics committee ap-proval was needed. A synopsis of the study protocol was, however, submitted to the local committee, which gave us the clearance to proceed.

Materials and Methods – the use of resources such as the Consolidated criteria for reporting qualitative research (COREQ): a 32-item checklist for interviews and focus groups is not described. What is the reason?

Thank you for your comment. We have added some items of the COREQ check list in the paper.

Discussion – it is suggested a greater depth in the discussion, even to support the Conclusion of the study.

Thank you for your comment. We have rewritten the discussion.

Reviewer 3 Report

Dear Authors

I read your paper with interest. Here is my suggestion, to improve the quality of your work.

Introduction

Clear and well-written. The aim of the study is stated.

Materials and Methods

Recruitment and sampling

Lines 57-61 reported results about students' sample selection. These lines should be moved to the results section.

Results

Please provide details about the meaning of letters and numbers between round brackets.

According to my review, the work requires minor revision.

Author Response

Lines 57-61 reported results about students' sample selection. These lines should be moved to the results section.

 Thank you for your comment. We have moved lines 57-61 to the results section.

Results

Please provide details about the meaning of letters and numbers between round brackets.

Thank you for your comment. The letters and numbers between round brackets are the ID of interviewed student

Round 2

Reviewer 2 Report

Accept in present form.